# Solar-to-hydrogen peroxide energy conversion on resorcinol–formaldehyde resin photocatalysts prepared by acid-catalysed polycondensation

Yasuhiro Shiraishi [1✉], Takumi Hagi[1], Masako Matsumoto[1], Shunsuke Tanaka [2], Satoshi Ichikawa[3] & Takayuki Hirai [1]

The photocatalytic generation of hydrogen peroxide from water and dioxygen ($H_2O + 1/2 O_2 \rightarrow H_2O_2$, $\Delta G° = +117\ kJ\ mol^{-1}$) under sunlight is a promising strategy for the artificial photosynthesis of a liquid fuel. We had previously found that resorcinol–formaldehyde (RF) resin powders prepared by the base-catalysed high-temperature hydrothermal method act as semiconductor photocatalysts for $H_2O_2$ generation. Herein, we report that RF resins prepared by the acid-catalysed high-temperature hydrothermal method (~523 K) using common acids at pH < 4 exhibit enhanced photocatalytic activity. The base- and acid-catalysed methods both produce methylene- and methine-bridged resins consisting of π-conjugated and π-stacked benzenoid–quinoid donor–acceptor resorcinol units. The acidic conditions result in the resins with a lower bandgap (1.7 eV) and higher conductivity because the lower-degree of crosslinking creates a strongly π-stacked architecture. The irradiation of the RF-acid resins with simulated sunlight in water with atmospheric-pressure $O_2$ generates $H_2O_2$ at a solar-to-chemical conversion efficiency of 0.7%, which is the highest efficiency ever reported for powder catalysts used in artificial photosynthesis.

[1] Research Center for Solar Energy Chemistry, and Division of Chemical Engineering, Graduate School of Engineering Science, Osaka University, Toyonaka 560-8531, Japan. [2] Department of Chemical, Energy and Environmental Engineering, Kansai University, Suita, Japan. [3] Research Center for Ultra-High Voltage Electron Microscopy, Osaka University, Ibaraki 567-0047, Japan. ✉email: shiraish@cheng.es.osaka-u.ac.jp

Artificial photosynthesis, which would allow for the transformation of earth-abundant resources into fuels, is a promising solar-driven technology for the realisation of a society based completely on sustainable energy[1]. The photocatalytic splitting of water to generate $H_2/O_2$[2,3], the reduction of $CO_2$ with water to CO or HCOOH[4,5], and the reduction of $N_2$ with water to $NH_3$[6,7] have been studied as methods for producing solar fuels, with particulate suspensions being the simplest systems for implementing these methods, owing to their ease of scalability[8,9]. In recent years, hydrogen peroxide ($H_2O_2$) has attracted significant attention as a solar fuel because it is a transportable liquid, can be used to generate electricity with a membrane-free one-compartment fuel cell[10], and can be readily produced from water and $O_2$ by semiconductor photocatalysis under ambient conditions[11,12]. The photogenerated valence band holes (VB $h^+$) are consumed by the oxidation of water (Eq. (1)) while the conduction band electrons (CB $e^-$) are consumed by the two-electron reduction of $O_2$ (Eq. (2)), resulting in the generation of $H_2O_2$ and a gain in free energy (Eq. (3)). Thus, the photocatalytic production of $H_2O_2$ is a highly promising technique for the artificial photosynthesis of a liquid solar fuel.

$$H_2O \rightleftharpoons 1/2\,O_2 + 2H^+ + 2e^- \,(+1.23\,V\,\text{vs. NHE, pH 0}) \quad (1)$$

$$O_2 + 2H^+ + 2e^- \rightleftharpoons H_2O_2\,(+0.68\,V\,\text{vs. NHE, pH 0}) \quad (2)$$

$$H_2O + 1/2\,O_2 \rightarrow H_2O_2\,(\Delta G^\circ = +117\,kJ\,mol^{-1}) \quad (3)$$

Several photocatalysts in the powdered form have been used for the generation of $H_2O_2$[13,14], but almost all of the systems suffer from low efficiency: the solar-to-chemical conversion (SCC) efficiency for the $H_2O_2$ generation on recently reported photocatalysts based on a graphitic carbon nitride (g-$C_3N_4$) are ~0.10% (g-$C_3N_4$/PDI: g-$C_3N_4$ doped with pyromellitic diimide)[15], ~0.20% (g-$C_3N_4$/PDI loaded with reduced graphene oxide)[16], ~0.26% (g-$C_3N_4$ reduced with $NaBH_4$)[17], and ~0.27% (g-$C_3N_4$/PDI loaded with reduced graphene oxide and boron nitride)[18], respectively. This is because they (i) do not absorb long-wavelength light ($\lambda > 500$ nm), (ii) show low activity with respect to the oxidation of water (Eq. (1)), (iii) show low selectivity with respect to the two-electron reduction of $O_2$ (Eq. (2)), and (iv) subsequently decompose the formed $H_2O_2$ by disproportionation or oxidation owing to the VB $h^+$ present on their surface (through the reverse reaction of that described in Eq. (2)). Thus, the development of semiconductor photocatalysts that efficiently catalyse the oxidation of water and the reduction of $O_2$ by absorbing long-wavelength light while exhibiting low activity for $H_2O_2$ decomposition is necessary.

Recently, we had demonstrated that resorcinol–formaldehyde (RF) resins[19], which have long been employed as "insulator" polymers[20–22], act as semiconductor photocatalysts for $H_2O_2$ generation through a simple synthesis[23]. We had found that the base-catalysed polycondensation of resorcinol with formaldehyde under high-temperature hydrothermal conditions (~523 K) affords methylene-bridged RF resin powders comprising the benzenoid forms of resorcinol, which are π-conjugated with the quinoid forms of resorcinol via methine linkers (Fig. 1a, b). This creates the resin particles consisting of the benzenoid–quinoid π-conjugated donor–acceptor (D–A) units (oxonol structure), which are linked to each other by non-π-conjugated methylene linkers. The charge-transfer (CT) interaction within the D–A unit creates small highest occupied molecular orbital–lowest unoccupied molecular orbital (HOMO–LUMO) gaps. The π-stacking of the D–A units (Fig. 1c) hybridises their energy levels and creates a low-gap (~2.0 eV) semiconducting band structure. The D and A units form the valence band (VB) and conduction band (CB), respectively; therefore, these units behave as active sites for

oxidation and reduction reactions[23]. The RF-base resins absorb long-wavelength light (~620 nm) and efficiently promote water oxidation (Eq. (1)) and $O_2$ reduction (Eq. (2)) while exhibiting low activity for the decomposition of the formed $H_2O_2$. The irradiation of the resins with simulated sunlight in water with $O_2$ produces $H_2O_2$ at a SCC efficiency of 0.5%, which is higher than that for powder catalysts during artificial photosynthesis reactions[1–7]. These low-bandgap polymers, which can be prepared using readily available and inexpensive reagents, may lead to the development of a route for the photocatalytic generation of $H_2O_2$.

The next challenge is increasing the activity of the resins. It is known that RF resins can also be prepared under acidic conditions, although all previously reported studies were performed at low temperatures (298–373 K)[20–22]. Both basic and acidic conditions result in methylene-bridged RF resins, albeit through different mechanisms:[22] the base-catalysed reaction is triggered by the deprotonation of resorcinol (Fig. 1a), whereas the acid-catalysed one is triggered by the protonation of formaldehyde (Fig. 2a). Therefore, the pH of the solution during the synthesis process affects the rate of condensation, the degree of crosslinking, composition, and morphology of the resins. Basic conditions usually yield highly crosslinked resins, whereas acidic conditions yield resins with a lower-degree of crosslinking[20–22]. In the present work, we synthesised RF resins under acidic conditions and used them for the photocatalytic generation of $H_2O_2$. Here, we report that the RF-acid resins prepared at pH levels lower than 4 and high-temperature hydrothermal conditions (~523 K) exhibit photocatalytic activity 1.5 times higher than that of RF-base resins, even though both types of resins are methylene-bridged resins containing π-conjugated and π-stacked D–A units. The RF-acid resins exhibit a lower bandgap (1.7 eV) and higher conductivity because their lower-degree of crosslinking leads to the creation of a strongly π-stacked D–A architecture owing to the higher structural flexibility. Further, the RF-acid resins produce $H_2O_2$ with an SCC efficiency of 0.7%, which is the highest ever for powder catalysts during artificial photosynthesis.

## Results and discussion

**Synthesis and morphology of resins.** The RF-acid resins were synthesised under hydrothermal conditions[23]. Each of the respective acids was added to a pure water containing resorcinol and formaldehyde (mole ratio of 1/2). The pH of the solution was adjusted to ~3 by controlling the amount of the acid added, unless noted otherwise. The formed reddish solution was kept in an autoclave at the designated temperature [$x$ (K)] for 24 h. The Soxhlet extraction of the solid using acetone and subsequent drying in vacuo afforded the RF-acid-$x$ resin powder. As shown in Fig. 3a, the RF-$(COOH)_2$-523 resin, which was prepared using oxalic acid (pH 3.0), is a red-black powder, whereas the RF-$NH_3$-523 resin, which was prepared by the base-catalysed hydrothermal method (pH 7.8), is a red-orange powder[23]. Scanning electron microscopy (SEM) images of RF-$(COOH)_2$-523 (Fig. 3b) indicated that its particles were spherical, as were those of the RF-base resins. Transmission electron microscopy (TEM) images of RF-$(COOH)_2$-523 (Fig. 3c) confirmed this result. Moreover, a series of tilted images of the resin powder (Supplementary Fig. 1) indicated that its particle exhibited the same shape and contrast even at different angles, thus further verifying the result. Dynamic light scattering (DLS) analysis (Supplementary Fig. 2) showed that the average diameter of the RF-$(COOH)_2$-523 particles (3.09 μm) was much larger than that of the RF-$NH_3$-523 particles (0.57 μm). $N_2$ adsorption/desorption analysis of RF-$(COOH)_2$-523 (Supplementary Fig. 3) indicated that it exhibited a Type III isotherm, as did RF-$NH_3$-523[23], suggesting the presence of

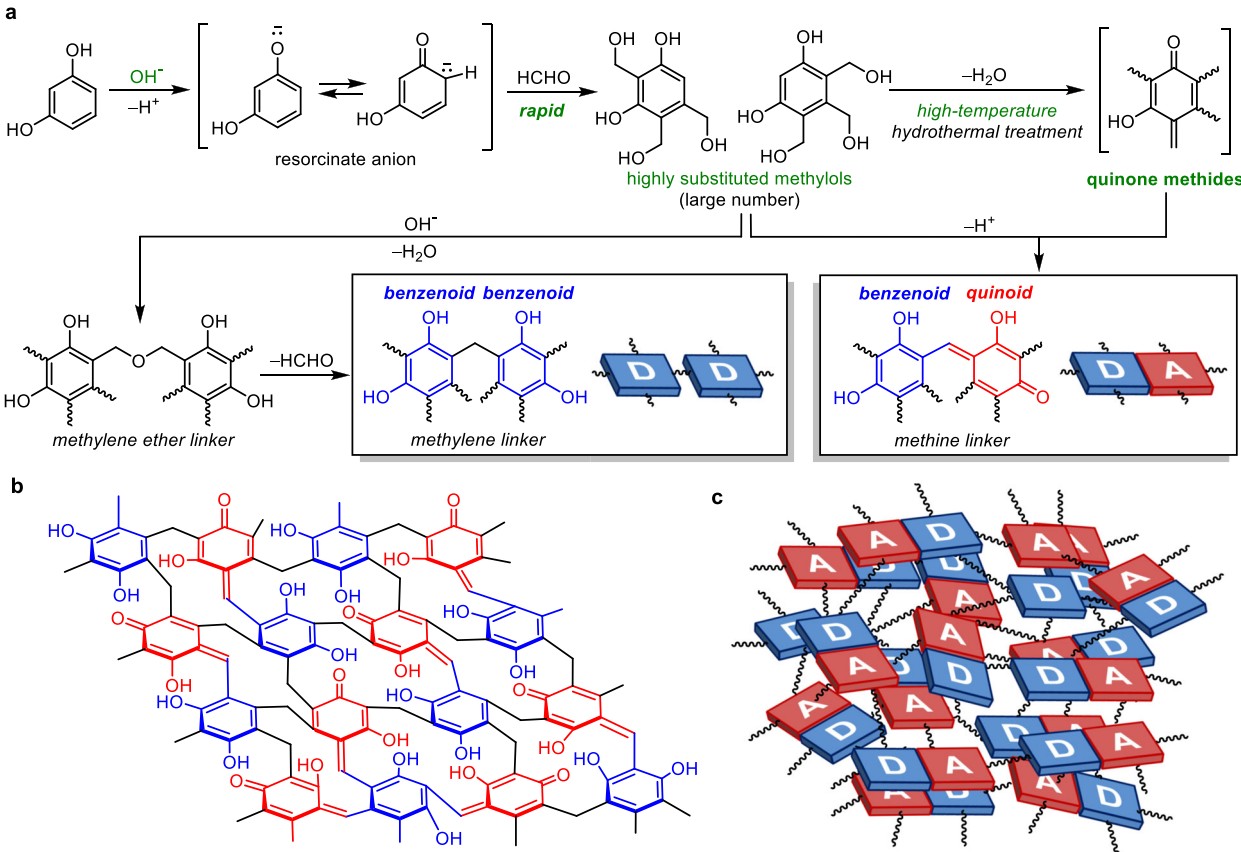

**Fig. 1 Polycondensation mechanism and structure of RF-base resins. a** Polycondensation mechanism; **b** crosslinking structure; and **c** π-conjugated and π-stacked D–A architecture. In **b**, all aromatic units are substituted by linkers. The structure contains 136 carbons (6 × 16 aromatics + 40 linkers), and the linker/aromatic ring ratio (= 40/16) is 2.5.

nonporous particles. The specific surface area of RF-$(COOH)_2$-523 (1.02 $m^2 g^{-1}$) was much smaller than that of RF-$NH_3$-523 (14.6 $m^2 g^{-1}$); this was in keeping with the particle sizes of the two resin powders. Moreover, as shown in Supplementary Fig. 4, the RF-acid-523 resin powders prepared at a pH of ~3 using other commercially available inorganic/organic acids (HCl, $H_2SO_4$, $HNO_3$, and $CH_3COOH$) also consisted of spherical particles with similar diameters (2.5–3.1 μm). These data indicated that, irrespective of the acid used, the acid-catalysed high-temperature hydrothermal method affords resins consisting of particles larger than those of RF-base resins.

**Photocatalytic activities of resins**. Photoreaction tests were performed under visible-light irradiation ($\lambda > 420$ nm) of a pure water (30 mL) containing the resin (50 mg) in an $O_2$ atmosphere (1 bar) at 298 K under magnetic stirring. A Xe lamp was used as the light source. Figure 4a shows the amounts of $H_2O_2$ generated on the different resins after 24 h of photoirradiation. The RF-$NH_3$-523 resin[23] produced ~60 μmol of $H_2O_2$ whereas the RF-acid-523 resins produced >79 μmol of $H_2O_2$, indicating that the RF-acid resins exhibited higher photocatalytic activity regardless of the acid used. The temperature during the hydrothermal process for resin synthesis must be high (523 K); it was observed that the activity of RF-$(COOH)_2$-x decreased with a decrease in the hydrothermal temperature. Changing the resorcinol/formaldehyde ratio and pH of the solution during the synthesis also creates spherical resin particles (Supplementary Fig. 5). The results of the photoreaction tests (Supplementary Table 1) confirmed that the acid-catalysed high-temperature hydrothermal process performed at 523 K using the resorcinol/formaldehyde

mole ratios in the range of 2/1–1/3 yielded resins with high photocatalytic activity. Figure 4b shows the relationship between the pH of the solution during the resin synthesis at 523 K with the 1/2 resorcinol/formaldehyde mole ratio and the amounts of $H_2O_2$ generated on the different resins after 24 h of photoirradiation. It can be seen clearly that, irrespective of the acid used, the activity of the resins was strongly affected by the pH during the synthesis, with the resins prepared at pH < 4 exhibiting very high activity for $H_2O_2$ generation.

**Compositions of resins**. As shown in Figs. 1b and 2b, the base- and acid-catalysed high-temperature hydrothermal processes yield methylene-bridged resins containing the benzenoid forms as electron donors (D), which are π-conjugated with the quinoid forms as electron acceptors (A) via methine linkers (oxonol structure). The formation of the quinoid units in the RF-acid resins was confirmed by infrared spectra (Supplementary Fig. 6). Both RF-$(COOH)_2$-523 and RF-$NH_3$-523 exhibited a band at 1650 $cm^{-1}$, assigned to the stretching of the C=O bonds of the quinoid units[23]. This was confirmed by the dipolar-decoupling magic-angle spinning $^{13}C$ nuclear magnetic resonance (DD/MAS/$^{13}C$ NMR) analysis. The spectra of both RF-$NH_3$-523 and RF-$(COOH)_2$-523 (Fig. 5) could be deconvoluted into the respective components (*a–o*)[24–27]. The resins exhibited a signal corresponding to the quinoid C=O carbon[28] at 182 ppm (*c*) as well as a signal corresponding to the methine carbon[29] at 126 ppm (*f*), with the areas being the same in both cases. This confirmed the formation of π-conjugated benzenoid–quinoid units (Fig. 2b). Table 1 shows the carbon compositions of the resins as determined by the integration of their $^{13}C$ NMR spectra

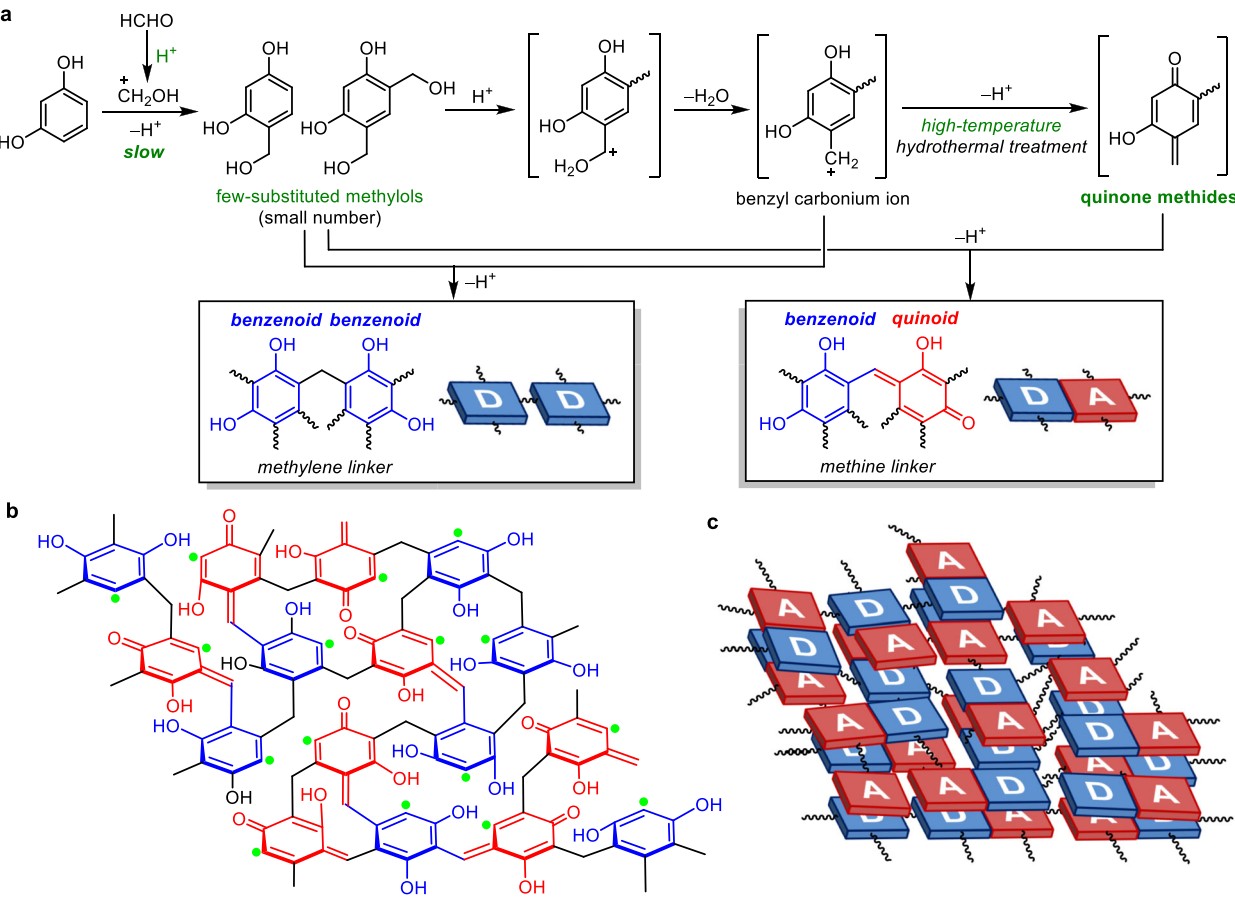

**Fig. 2 Polycondensation mechanism and structure of RF-acid resins. a** Polycondensation mechanism; **b** bridging structure; and **c** π-conjugated and π-stacked D–A architecture. In **b**, the green dots denote the sites not substituted by linkers. The structure contains 126 carbons (6 × 16 aromatics + 30 linkers), and the linker/aromatic ring ratio (= 30/16) is 1.88.

(Supplementary Fig. 7). The X-ray photoelectron spectroscopy (XPS) analyses of the resins at the C 1s (Supplementary Fig. 8) and O 1s levels (Supplementary Fig. 9) confirmed the presence of C–O and C=O bonds[30]. The C–O/C=O ratios of the resins as determined from the O 1s spectra were in keeping with those determined from the $^{13}C$ NMR spectra (Supplementary Table 2), thus verifying that the determined compositions were accurate (Table 1). The benzenoid/quinoid ratio of RF-(COOH)$_2$-523 is 51/49 and similar to that of RF-NH$_3$-523 (54/46)[23]. Thus, both resins contain similar numbers of D and A units.

**Degrees of crosslinking of resins**. Table 1 shows that the RF-base and RF-acid resins both contain methylene and methine groups as the main linkers along with similar numbers of residual groups (~10%). The shapes of their NMR spectra (Fig. 5) indicated that both resins contain fewer linker carbons (***l*–*n***) than aromatic carbons (***d*–*i***), with the proportion of the linker carbons in RF-(COOH)$_2$-523 being lower than that in RF-NH$_3$-523. Further, as can also be seen from Table 1, the total proportion of the linker carbons in RF-(COOH)$_2$-523 is 23.5%, which is much lower than that in RF-NH$_3$-523 (31.2%). The ratio of the number of the linkers to the number of aromatic ring was determined to be ~2 for RF-(COOH)$_2$-523 and ~3 for RF-NH$_3$-523. The average number of linkers substituted in a single aromatic ring was ~3 for RF-(COOH)$_2$-523 and ~4 for RF-NH$_3$-523 because each linker bridges two aromatic rings. This indicated that, in the case of the RF-base resins (Fig. 1b), all the four unsubstituted sites in resorcinol (benzenoid and quinoid) are completely substituted by the methylene or methine linkers, leading to the formation of a highly crosslinked structure. In contrast, in the case of the RF-acid resins (Fig. 2b), three of the four unsubstituted sites in resorcinol are substituted by the linkers, resulting in a lower-degree-crosslinked structure, as was also observed in previous studies on RF resins prepared under acidic conditions[20–22].

**Mechanisms of resin formation**. The different degrees of crosslinking of the RF-base and RF-acid resins originate from the differences in their condensation mechanisms[20–22]. The base-catalysed condensation (Fig. 1a) is initiated by the deprotonation of resorcinol. The rapid hydroxymethylation of the resorcinate anions by formaldehyde results in a large number of highly substituted methylols. Their condensation produces methylene ether linkers, and the subsequent decomposition produces methylene linkers, resulting in highly crosslinked resins[22]. In addition, the high-temperature hydrothermal process removes the –OH groups from the methylols, leading to the formation of highly electrophilic quinone methides[31,32]. The nucleophilic addition of the methylols to the quinone methides[33] produces quinoid forms, which are π-conjugated with benzenoid forms. These reactions create a highly crosslinked D–A architecture (Fig. 1b). In contrast, the acid-catalysed condensation (Fig. 2a) is triggered by the protonation of formaldehyde, which promotes hydroxymethylation of resorcinol. The protonation of the methylols and the subsequent dehydration produce benzyl carbonium ions, which reacts with methylols, leading to the formation of methylene linkers. In this case, slow hydroxymethylation produces a smaller number of few-substituted methylols and benzyl carbonium ions, thus resulting in the

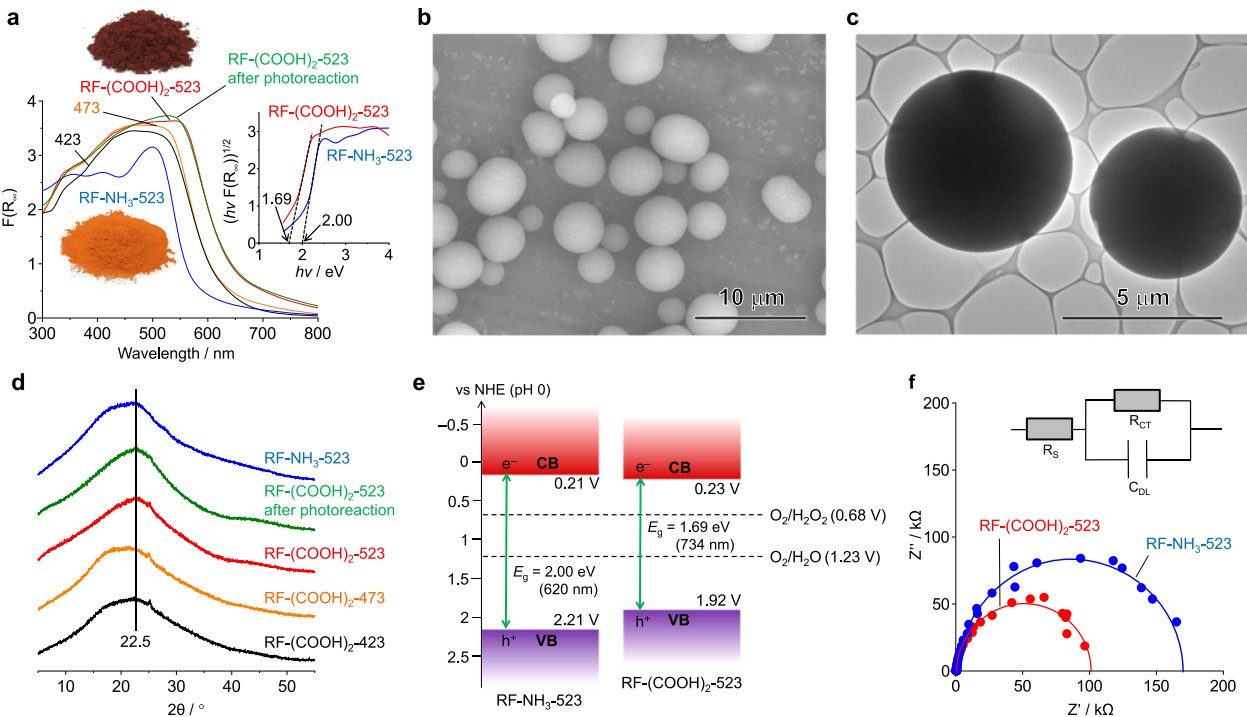

**Fig. 3 Electronic and structural properties of RF resins. a** DR UV–Vis spectra of various resins and that of RF-(COOH)$_2$-523 resin recovered after photoreaction for 5 h under solar simulator (Fig. 4d, red square); inset shows Tauc plots. **b** SEM and **c** TEM micrographs of RF-(COOH)$_2$-523 resin. **d** XRD patterns and **e** electronic band structures of the resins. **f** EIS Nyquist plots of the resins measured in 0.1 M KCl under irradiation with visible-light at bias of 0.8 V (vs. Ag/AgCl) for frequencies of 10 mHz to 10 kHz. Equivalent circuit (inset) contains ohmic resistance ($R_S$), double-layer capacitance ($C_{DL}$), and charge-transfer resistance ($R_{CT}$).

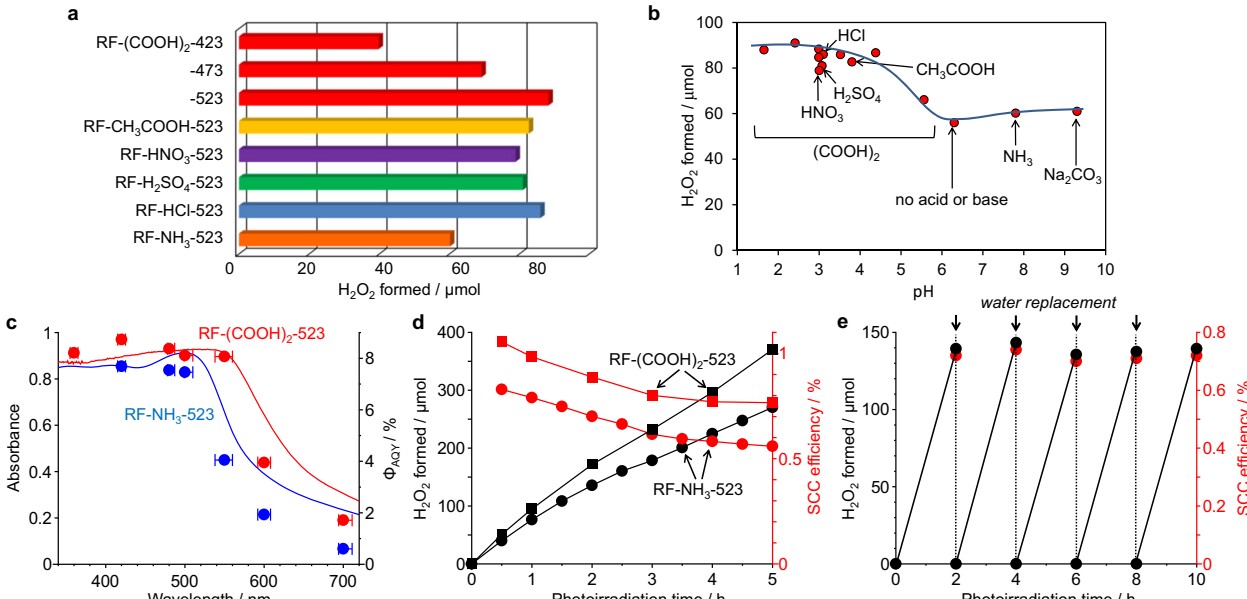

**Fig. 4 Photocatalytic properties of RF resins. a** Amounts of H$_2$O$_2$ generated on various resins after 24 h of photoirradiation. Conditions: water (30 mL), catalyst (50 mg), O$_2$ (1 bar), $\lambda$ > 420 nm (Xe lamp, light intensity at 420−700 nm: 140.3 W m$^{-2}$), temperature (298 K). All of the data are the mean values determined by three independent experiments and contain ±6% deviations. **b** Relationship between pH of solution during resin synthesis at 523 K with the resorcinol/formaldehyde mole ratio of 1/2 and amounts of H$_2$O$_2$ generated on various resins after 24 h of photoirradiation. **c** Absorption spectra of resins and action spectra for H$_2$O$_2$ generation on resins; $\Phi_{AQY}$ values were determined using Eq. (5). **d** Changes in amounts of H$_2$O$_2$ generated on resins and corresponding SCC efficiency under AM1.5 G simulated sunlight (1 sun). Conditions: water (50 mL), catalyst (250 mg), O$_2$ (1 bar), $\lambda$ > 420 nm (solar simulator, light intensity at 420–700 nm: 410.0 W m$^{-2}$), temperature (333 K). **e** Results of repeated photoreaction tests for RF-(COOH)$_2$-523 under simulated sunlight irradiation. After 5 h of photoreaction (**d**), resin was recovered by centrifugation and used again for photoirradiation tests; water sample was replaced every 2 h.

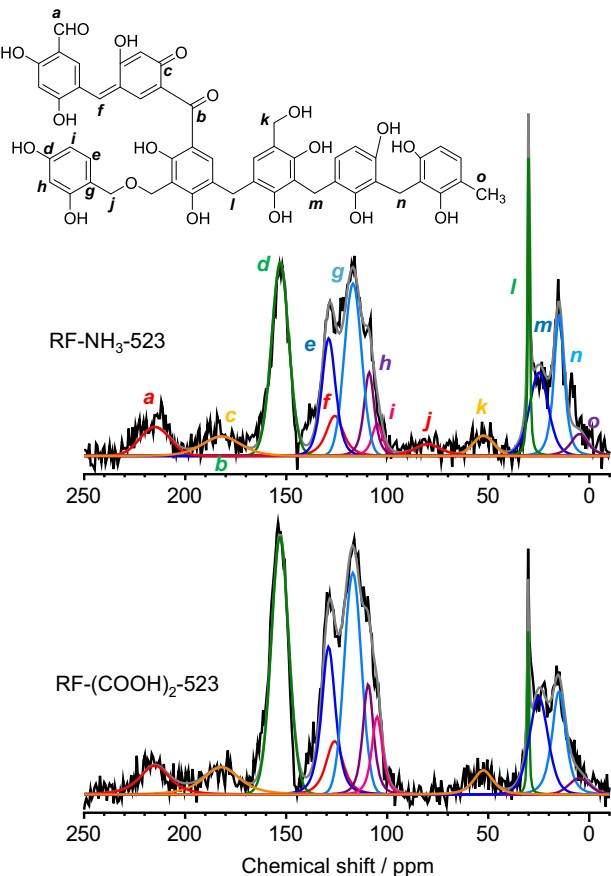

RF-NH$_3$-523

RF-(COOH)$_2$-523

Chemical shift / ppm

**Fig. 5 Solid-state DD/MAS/$^{13}$C NMR spectra of RF-NH$_3$-523 and RF-(COOH)$_2$-523 resins.** Assignments of carbon components: aldehyde –CHO (210 ppm: **a**), ketone C=O (190 ppm: **b**), quinone C=O (182 ppm: **c**), resorcinol C–OH (153 ppm: **d**), nonsubstituted resorcinol C at *meta* position (130 ppm: **e**), methine linker –C= (126 ppm: **f**), substituted resorcinol C (117 ppm: **g**), nonsubstituted resorcinol C at *para* and *ortho* positions (110 ppm: **h**, 105 ppm: **i**), methylene ether linker –C–O–C– (70 ppm: **j**), methylol C–OH (55 ppm: **k**), methylene linker –C– substituted to 4,4′-, 2,4′-, and 2,2′-positions of resorcinol (30 ppm: l, 20 ppm: **m**, 10 ppm: **n**), and methyl –CH$_3$ (5 ppm: **o**).

formation of lower-degree-crosslinked resins[22]. In addition, the high-temperature hydrothermal process promotes the deprotonation of the benzyl carbonium ions, which results in quinone methides[34], leading to the formation of quinoid units. These reactions create a lower-degree-crosslinked D–A architecture (Fig. 2b). The rapid and slow crosslinking under basic and acidic conditions, respectively, was confirmed by the sizes of the resin particles formed. The resins form via the nucleation/growth (LaMer) mechanism[35]. Under basic conditions (Fig. 1a), the rapid hydroxymethylation and condensation produce a large number of nuclei, resulting in the formation of small resin particles. Under acidic conditions (Fig. 2a), the slow hydroxymethylation inevitably produces fewer nuclei and promotes their gradual growth, resulting in the formation of larger resin particles. As shown in Supplementary Fig. 1, the size of the resin particles did increase with a decrease in the pH during the synthesis. This is in keeping with the proposed mechanism, namely, that the crosslinking process is slower under acidic conditions (Fig. 2a) and results in the formation of a lower-degree-crosslinked architecture (Fig. 2b).

**Π-stacking in resins**. The CT interactions within the formed D–A unit lead to the formation of a small HOMO–LUMO gap. The π-

**Table 1 Carbon compositions of various RF resins.**

| Resin | Aromatics/% (*c, d, e, g, h, i*) | Linkers/% | | | | Residual groups/% | | | Benzenoid /quinoid[a] | Linker /aromatic ring[b] |
|---|---|---|---|---|---|---|---|---|---|---|
| | | Methylene ether (*j*) | Methylene (*l, m, n*) | Methine (*f*) | Ketone (*b*) | Aldehyde (*a*) | Methylol (*k*) | Methyl (*o*) | | |
| RF-NH$_3$-523 | 59.8 | 2.1 | 24.5 | 4.6 | 0 | 5.0 | 2.4 | 1.7 | 54/46 | 3.02 |
| RF-(COOH)$_2$-523 | 68.1 | 0 | 18.5 | 5.0 | 0 | 4.7 | 2.1 | 1.6 | 51/49 | 2.07 |
| RF-(COOH)$_2$-523$^c$ | 68.0 | 0 | 14.1 | 5.1 | 4.5 | 5.5 | 1.4 | 1.5 | 50/50 | 2.09 |
| RF-(COOH)$_2$-473 | 67.3 | 0 | 18.3 | 4.3 | 0 | 6.2 | 2.7 | 1.2 | 62/38 | 2.01 |
| RF-(COOH)$_2$-423 | 68.9 | 0 | 16.2 | 3.5 | 0 | 7.2 | 3.3 | 0.8 | 69/31 | 1.71 |

The compositions were determined by integration of solid-state DD/MAS/$^{13}$C NMR spectra (Fig. 5 and Supplementary Fig. 7). The letters in parentheses denote the corresponding carbons shown in the resin structure (Fig. 5).
[a]Mole fractions of quinoid and benzenoid units in resins were determined from respective peak areas using following equations:

$$\%quinoid = \left( \frac{[quinone\ C=O\ carbon(\mathbf{c})] \times 6 \times 100 / [all\ aromatic\ carbons\ (\mathbf{c},\mathbf{d},\mathbf{e},\mathbf{g},\mathbf{h},\mathbf{i})]}{[all\ carbons(\mathbf{a}-\mathbf{o})]} \right) \times 100$$

$$\%benzenoid = 100 - \%quinoid$$

[b]Ratio of number of linkers to number of aromatic rings in resins was determined using following equation: linker /aromatic ring = $\frac{[all\ linker\ carbons\ (\mathbf{b},\mathbf{f},\mathbf{j}/2,\mathbf{l},\mathbf{m},\mathbf{n})]}{[all\ aromatic\ carbons\ (\mathbf{c},\mathbf{d},\mathbf{e},\mathbf{g},\mathbf{h},\mathbf{i})]} \times 6$

[c]RF-(COOH)$_2$-523 resin recovered after photoreaction for 5h in water with O$_2$ using solar simulator (Fig. 4d).

stacking of the D–A units hybridises the energy levels and creates a semiconducting band structure[23]. Under the basic conditions, highly crosslinked rigid resins are formed, wherein the D–A units exhibit a weaker π-stacking owing to the structural restrictions (Fig. 1c). In contrast, under the acidic conditions, lower-degree-crosslinked resins are formed, wherein the D–A units exhibit a stronger π-stacking because of the structural flexibility (Fig. 2c). The stronger π-stacking in the RF-acid resins enhances the delocalisation of the π-electrons and creates the resins with lower bandgap and higher conductivity. This is the reason they exhibit enhanced photocatalytic activity (Fig. 4a). The stronger π-stacking of the RF-acid resins was confirmed through powder X-ray diffraction (XRD) analysis (Fig. 3d). The resins exhibited a broad diffraction peak at $2\theta = \sim 20°$ ($d = \sim 4.4$ Å), which could be assigned to π-stacked aromatics. Moreover, the peak of RF-(COOH)$_2$-523 was shifted to higher angles with respect to those of RF-NH$_3$-523. This indicated that the D–A distance in RF-(COOH)$_2$-523 was smaller, even though both resins contained the similar numbers of D and A units (Table 1). The RF-acid-523 resins prepared using other inorganic/organic acids (HCl, H$_2$SO$_4$, HNO$_3$, and CH$_3$COOH) also showed peaks that were shifted to higher angles (Supplementary Fig. 10). The acid-catalysed synthesis yields resins with strongly π-stacked D–A units via the formation of a lower-degree-crosslinked architecture (Fig. 2c).

**Electronic properties of resins**. The diffuse-reflectance (DR) ultraviolet–visible (UV–vis) spectra of the resins (Fig. 3a) show absorption bands in the visible region; these were related to the CT transitions of the π-conjugated and π-stacked D–A units. RF-(COOH)$_2$-523 showed stronger band in the longer-wavelength region as compared with that of RF-NH$_3$-523, confirming stronger π-stacking of the D–A units[36]. Further, the RF-acid-523 resins prepared using the other acids also exhibited stronger absorption (Supplementary Fig. 10). It is noted that, under the present high-temperature conditions for the resin synthesis (>473 K), a dehydration condensation of –OH groups at the 2,2′ positions of the D–A units (Figs. 1a and 2a, right bottom) may also proceed to form a 6-hydroxy-3-fluorone moiety involved in xanthene dyes[37], although their formation is not confirmed at present. This is because these moieties usually show maximum absorption at ~520 nm[38] similar to the absorption spectra of the resins. The Mott-Schottky plot of RF-(COOH)$_2$-523 has a positive slope (Supplementary Fig. 11), indicating that the resin acts as an n-type semiconductor, just like the RF-base resins[23]. Further, the potential and bandgap (1.69 eV) of RF-(COOH)$_2$-523 as determined from its Tauc plot (Fig. 3a, inset) were used to determine its band structure (Fig. 3e). The bottom of its CB (+0.23 V versus RHE) is lower than that of RF-NH$_3$-523 (+0.21 V) while the top of its VB (+1.92 V versus RHE) is higher than that of RF-NH$_3$-523 (+2.21 V). In addition, the CB and VB levels of RF-(COOH)$_2$-523 are sufficient for the reduction of O$_2$ (+0.68 V, Eq. (3)) and the oxidation of water (+1.23 V, Eq. (2)). The results of electron spin resonance (ESR) analysis of the resins (Supplementary Fig. 12) showed a Lorentzian line corresponding to the π-electrons ($g = 2.005$)[39]. The intensity in the case of RF-(COOH)$_2$-523 was stronger than that of RF-NH$_3$-523, indicating enhanced π-electron delocalisation in the former case. Moreover, the photoirradiation of the resin increased the intensity, suggesting that the photoexcited resin generated h$^+$–e$^-$ pairs. The irradiation of RF-(COOH)$_2$-523 with $\lambda > 420$ nm light or with monochromated 550-nm light on a fluorine tin oxide (FTO) electrode resulted in a stronger photocurrent than that seen in the case of RF-NH$_3$-523 (Supplementary Fig. 13), thus confirming enhanced h$^+$–e$^-$ generation in the former case. Further, the results of electrochemical impedance spectroscopy (EIS) performed under visible-light (Fig. 3f) indicated that RF-(COOH)$_2$-

523 had a lower CT resistance ($R_{CT}$) than that of RF-NH$_3$-523, indicating that the increased conductivity of RF-(COOH)$_2$-523 enhances migration of the CB e$^-$ photogenerated in the resin. These findings suggest that strongly π-stacked D–A systems (Fig. 2c) result in low-bandgap, highly conductive resins that efficiently generate h$^+$–e$^-$ pairs under visible-light irradiation.

**Effects of temperature during hydrothermal process**. The temperature during the hydrothermal process (~523 K) for the synthesis of the RF-acid resins must be high, as lower temperatures decrease the photocatalytic activity of the resins (Fig. 4a). As shown in Table 1, the benzenoid/quinoid ratios of the RF-(COOH)$_2$-423 and −473 resins were 69/31 and 62/38, respectively; thus, these resins contained fewer quinoid units than RF-(COOH)$_2$-523 (51/49). Accordingly, the π-stacking-related diffraction peaks of these resins (Fig. 3d) appeared at lower angles than those of RF-(COOH)$_2$-523, indicating that the RF-acid resins prepared at lower hydrothermal temperatures contained fewer quinoid units. The high-temperature, high-pressure conditions probably enhance the deprotonation of the benzyl carbonium ions (Fig. 2a) and produce a larger number of quinone methides[40,41], thus resulting in more quinoid units. These data indicate that the acid-catalysed high-temperature hydrothermal synthesis method leads to a lower-degree-crosslinked structure while also producing a large number of quinoid units. The flexibility of this structure facilitates strong D–A π-stacking (Fig. 2c), resulting in lower bandgap, highly conductive resins.

**Photocatalytic properties of resins**. The action spectrum for H$_2$O$_2$ generation on RF-(COOH)$_2$-523 (Fig. 4c) indicated that the apparent quantum yield ($\Phi_{AQY}$) of the resin was in keeping with its absorption spectrum, meaning that the bandgap photoexcitation leads to the H$_2$O$_2$ generation. The resin showed a larger $\Phi_{AQY}$ value than that of RF-NH$_3$-523 over the entire wavelength region, representing its broader, stronger light absorption and higher conductivity. The photoexcited resin catalyses the reduction of O$_2$ through the CB e$^-$ (Eq. (3)) and the oxidation of water through the VB h$^+$ (Eq. (2)), as confirmed by the following reactions. The photoirradiation of RF-(COOH)$_2$-523 in water with benzyl alcohol as an electron donor[42] under O$_2$ (Supplementary Fig. 14) produced H$_2$O$_2$, with the activity being higher than that of RF-NH$_3$-523. Further, the photoirradiation of RF-(COOH)$_2$-523 in water with NaIO$_3$ as an electron acceptor[43] under Ar (Supplementary Fig. 15) produced O$_2$, with the activity being higher than that of RF-NH$_3$-523. In this case, H$_2$O$_2$ did not form during the reaction, meaning that the two-electron oxidation of water by the VB h$^+$ (Eq. (4))[44] did not occur. The electrochemical linear sweep voltammetry (LSV) curves of the resins (Supplementary Fig. 16) indicated a cathodic current at < +0.2 V (versus RHE) for O$_2$ reduction (Eq. (2)) and an anodic current at > +1.7 V for water oxidation (Eq. (1)). Further, the cathodic and anodic currents of RF-(COOH)$_2$-523 were greater than those of RF-NH$_3$-523. These data confirmed that the oxidation of water and reduction of O$_2$ occurred efficiently on the RF-(COOH)$_2$-523 resin by the photogenerated h$^+$–e$^-$ pairs.

$$2H_2O \rightarrow H_2O_2 + 2H^+ + 2e^- \; (+1.77 \, V \, vs. \, NHE) \qquad (4)$$

Supplementary Fig. 17 shows the results of photoirradiation of RF-(COOH)$_2$-523 in water containing H$_2$O$_2$ and NaIO$_3$ as an electron acceptor under Ar. The activity for oxidative decomposition of H$_2$O$_2$ by the VB h$^+$ on the catalyst (the reverse reaction in Eq. (2)) is similar to that on the RF-NH$_3$-523[23]. This indicates that the RF-acid resins are also less active for H$_2$O$_2$ decomposition and contribute to efficient H$_2$O$_2$ production.

**Artificial photosynthesis performances of resins**. The activities of the RF-acid resins were evaluated under illumination with AM1.5 G simulated sunlight (1 sun)[45] (Supplementary Fig. 18). Figure 4d shows the change in the amount of $H_2O_2$ generated with time. The rate of $H_2O_2$ formed on RF-$(COOH)_2$-523 is higher than that on RF-$NH_3$-523, again confirming that the activity of the former was higher. It can also be observed that, in the early stage, the formation rate increases rapidly but eventually becomes constant after prolonged photoirradiation (~3 h), as is also the case for RF-$NH_3$-523[23]. Accordingly, the SCC efficiency for $H_2O_2$ formation becomes almost constant at ~0.7%, which is higher than that of RF-$NH_3$-523 (~0.5%). The high activity in the early stage is attributable to the fact that the formed VB $h^+$ are consumed by the self-oxidation of the resin in addition to the oxidation of water. This was also observed in the case of the RF-base resins[23]. Supplementary Fig. 7 shows the NMR spectrum of the resin after photoirradiation for 5 h in water with $O_2$ (Fig. 4d). As shown in Table 1, the photoirradiation decreased the numbers of methylol (***k***) and methylene (***l***, ***m***, ***n***) carbons but increased the numbers of aldehyde (***a***) and ketone (***b***) carbons, indicating that the methylol and methylene parts of the resin are oxidised by the VB $h^+$, as is also the case for the RF-base resins[23]. However, its benzenoid/quinoid ratio (51/49) did not change noticeably after the reaction (50/50). Moreover, the absorption spectrum and XRD pattern of the recovered resin were similar to those in the fresh state (Fig. 3a and d). SEM observations of the recovered resin (Supplementary Fig. 19) confirmed the presence of spherical particles (3.15 μm), which were similar to those observed in the resin in the fresh state both in terms of shape and size (3.09 μm) (Supplementary Fig. 2). In addition, TEM observations of the recovered resin confirmed the presence of round particles (Supplementary Fig. 20), and a series of tilting images of an individual particle confirmed their shape (Supplementary Fig. 21). These data indicated that the structure, morphology, and semiconducting properties of the resin did not change after the photoirradiation. As shown in Fig. 4e, even after being reused five times, the recovered resin maintained an SCC efficiency of 0.7% for $H_2O_2$ generation without any significant loss in activity. This is higher than the solar-to-biomass conversion efficiency of the natural photosynthesis in typical plants (~0.1%)[46] and the highest value ever reported for photocatalytic water splitting using powder catalysts (~0.5%)[47]. Thus, the stable generation of $H_2O_2$ on the RF-acid resins under sunlight has significant potential for use in the production of a liquid solar fuel.

## Conclusion

As stated above, we found that RF-acid resins prepared by the acid-catalysed hydrothermal synthesis method (~523 K) act as highly active semiconductors for the photocatalytic production of $H_2O_2$ in water in the presence of atmospheric-pressure $O_2$ under sunlight irradiation. Further, synthesis at lower pH levels (<4) results in a lower-degree-crosslinked RF resins with a strongly π-stacked benzenoid–quinoid D–A architecture. This, in turn, results in highly conductive, lower bandgap (~1.7 eV) resins that efficiently promote water oxidation and $O_2$ reduction under irradiation with near-infrared (λ < 720 nm) light. These RF-acid resins generate $H_2O_2$ with an SCC efficiency of more than 0.7%, which is the highest ever for artificial photosynthesis using powder photocatalysts. Further, these resins can be prepared through a simple hydrothermal reaction using inexpensive reagents (resorcinol, formaldehyde, acids, and water). The results reported here may, therefore, contribute to the development of low-bandgap semiconducting polymers and highly active photocatalysts for $H_2O_2$ generation, which, in turn, may aid the production of liquid solar fuels.

## Methods

**Materials**. The RF-acid-523 resins (acid = HCl, $H_2SO_4$, $HNO_3$, $(COOH)_2$, or $CH_3COOH$) were prepared as follows. To being with, resorcinol (400 mg, 3.6 mmol) and formaldehyde (33 wt% solution, 540 μL, 7.2 mmol) were added to water (40 mL) under stirring at room temperature. Next, the acid in question was added to the solution, and the pH of the solution was adjusted to ~3. The solution was then transferred to an autoclave (capacity: 100 mL) and heated for 24 h (heating rate: 7 K $min^{-1}$), where the cooling operation of the autoclave was done by natural air cooling. The resultant product was subjected to Soxhlet extraction using acetone (12 h) and dried in vacuo at room temperature (12 h).

**Photoreaction**. The catalyst (50 mg) being tested and water (30 mL) were added to a glass bottle (φ: 35 mm; capacity: 50 mL), which was then sealed using a rubber cap. Next, $O_2$ was bubbled into the bottle, which was then photoirradiated under stirring with a magnetic stirrer using light with λ > 420 nm from a 2-kW Xe lamp[15]. The $\Phi_{AQY}$ value corresponding to the reaction at 298 K was calculated using the following equation:

$$\Phi_{AQY}(\%) = \frac{[\text{Amount of } H_2O_2 \text{ generated(mol)}] \times 2}{[\text{Number of photons entering the reactor(mol)}]} \times 100 \quad (5)$$

The SCC efficiency was determined using the following equation[16]:

$$\text{SCC efficiency}(\%) = \frac{[\Delta G \text{ for } H_2O_2 \text{ generation(J mol}^{-1})] \times [\text{Amount of } H_2O_2 \text{ formed(mol)}]}{[\text{total input energy(W)}] \times [\text{reaction time(s)}]} \times 100 \quad (6)$$

The free energy for $H_2O_2$ generation is 117 kJ $mol^{-1}$ (Eq. (4)) while the total input energy was 0.314 W[23]. The amount of $H_2O_2$ generated was quantified using an HPLC system equipped with an electrochemical analyser[23]. The amounts of $H_2O_2$ in all of the samples are the mean values determined by three analysis and the values contain only ±2% deviations.

**Analysis**. DD/MAS/$^{13}$C NMR[23], ESR (X band), DR UV–Vis, XRD, and XPS analyses (Mg-Kα radiation)[48] and SEM[49] and TEM imaging[50] were performed as per previously reported procedures. The electrochemical measurements were performed using 0.1 M $Na_2SO_4$ (pH 6.6) as the electrolyte and a Pt wire and an Ag/AgCl electrode as the counter and reference electrodes, respectively[50]. The working electrodes were prepared with FTO, carbon paper, or indium tin oxide[23]. The LSV measurements were performed in an $O_2$- or Ar-saturated 0.5 M phosphate buffer solution (pH 7)[51]. All the potential values were expressed with respect to the RHE for ease of comparison of the data obtained at different pH values. This was done using the following equations[52]:

$$E(\text{vs.RHE}) = E(\text{vs.Ag/AgCl}) + E_{Ag/AgCl}(\text{ref}) + 0.0591\ V \times \text{pH}\left(E_{Ag/AgCl}(\text{ref})\right)$$
$$= 0.1976\ V \text{ vs.NHE at 25 ° C)} \quad (7)$$

## Data availability

All experimental data within the article and its Supplementary Information are available from the corresponding author upon reasonable request.

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

## Acknowledgements

This work was supported by the Precursory Research for Embryonic Science and Technology (PRESTO) program of the Japan Science and Technology Agency (JST) and a Grant-in-Aid for Scientific Research on Innovative Areas (No. 20H05100) from the Ministry of Education, Culture, Sports, Science and Technology, Japan (MEXT).

## Author contributions

Y.S. directed this project; T.H., M.M., and T.H. performed the experiments and analysed the data; S.T. performed the XPS analysis; and S.I. performed the TEM observations. The manuscript was written by Y.S. with contributions from all the other authors.

## Competing interests

The authors declare no competing interests.
