## [Peer Review File · Communications Chemistry]

This manuscript has been previously reviewed at another Nature Research journal. This document only contains reviewer comments and rebuttal letters for versions considered at Communications Chemistry.

REVIEWERS' COMMENTS:

Reviewer #1 (Remarks to the Author):

The authors replied satisfactorily to the queries raised by the reviewers. It is my opinion that the revised manuscript meet the criteria for being published as it is.

Reviewer #2 (Remarks to the Author):

The previous comments have been well addressed. I am happy to recommend a publication in NC.

Reviewer #3 (Remarks to the Author):

I do not like to repeat my general description concerning this manuscript (compared to the previous submission to Nature Communications). There are two aspects should be evaluated in this case. The first one is general importance of the results. After reevaluating the manuscript I came to the conclusion that the level of importance of the results fits the vision of Communications Chemistry editors.

The second aspect is the description of the catalyst. The authors wrote the rebuttal, but I can not fully accept their arguments. Regardless of the fact if C-O-C bond is formed or not, the dyes being product of condensation of formaldehyde and resorcinol and possessing absorption at 500 nm have to possess oxonol-type structure. Either it is cyclic or not is debatable. One has to just compare the published abs.max for 6-hydroxy-3-fluorone (JOC, 1992, 4418 – 520 nm) with the one reported herein (~525 nm), to see the similarity. Examining the papers on the condensation of formaldehyde with resorcinol (Berichte, 25, 947, Berichte. 27, 2888, JACS 1932, 54, 4325) is inconclusive because they were published at the times when analytical techniques were limited. On the other hand other aldehydes usually give fluorone type product in the presence of oxidants. At minimum authors should present second structural possibility (i.e. fluorone ring closure) as hypothesis.

I completely disagree that oxonols supposed to have more methine carbons (than one).

Fluorescein (possessing one methine bridge) belongs to oxonols. Oxonols have one feature – the structure is delocalized, quinoid and phenolic parts 'are hybridized'. As authors presented of Fig. 2b these are polymethine dyes linked together with insulating CH₂ linkers. The structure of this type of polymeric material will never be entirely clear also because it probably contain 'various moieties' i.e. it is not homogenous. In this particular case the performance of the material is more important than the structure. Therefor on the balance I recommending for acceptance, but I urge authors to add another point of view in their text using sentence such as:

'The presence of 6-hydroxy-3-fluorone moieties in the polymeric structure can not be excluded especially in the material products under acidic conditions.'

'The absorption spectrum of our materials correspond to 6-hydroxy-3-fluorone.citation'

I recommend the acceptance conditioning that authors modify their description of catalysts' structure, following above-described arguments.

Response to Comments

Manuscript ID: COMMSCHEM-20-0329-T

Title: Solar-to-hydrogen peroxide energy conversion on resorcinol-formaldehyde resin photocatalysts prepared by acid-catalysed polycondensation

Authors: Yasuhiro Shiraishi,* Takumi Hagi, Masako Matsumoto, Shunsuke Tanaka, Satoshi Ichikawa, and Takayuki Hirai

NOTE: The revised parts in the manuscript are indicated by **YELLOW** background.

Answers to comments by Reviewer #3

(Comment) *I do not like to repeat my general description concerning this manuscript (compared to the previous submission to Nature Communications). There are two aspects should be evaluated in this case. The first one is general importance of the results. After reevaluating the manuscript I came to the conclusion that the level of importance of the results fits the vision of Communications Chemistry editors.*

The second aspect is the description of the catalyst. The authors wrote the rebuttal, but I can not fully accept their arguments. Regardless of the fact if C-O-C bond is formed or not, the dyes being product of condensation of formaldehyde and resorcinol and possessing absorption at 500 nm have to possess oxonol-type structure. Either it is cyclic or not is debatable. One has to just compare the published abs.max for 6-hydroxy-3-fluorone (JOC, 1992, 4418 – 520 nm) with the one reported herein (~525 nm), to see the similarity. Examining the papers on the condensation of formaldehyde with resorcinol (Berichte, 25, 947, Berichte. 27, 2888, JACS 1932, 54, 4325) is inconclusive because they were published at the times when analytical techniques were limited. On the other hand other aldehydes usually give fluorone type product in the presence of oxidants. At minimum authors should present second structural possibility (i.e. fluorone ring closure) as hypothesis.

I completely disagree that oxonols supposed to have more methine carbons (than one). Fluorescein (possessing one methine bridge) belongs to oxonols. Oxonols have one feature – the structure is delocalized, quinoid and phenolic parts ‘are hybridized’. As authors presented of Fig. 2b these are polymethine dyes linked together with insulating CH₂ linkers. The structure of this type of polymeric material will never be entirely clear also because it probably contain ‘various moieties’ i.e. it is not homogenous. In this particular case the performance of the material is more important than the structure. Therefore on the balance I recommending for acceptance, but I urge authors to add another point of view in their text using sentence such as:

‘The presence of 6-hydroxy-3-fluorone moieties in the polymeric structure can not be excluded especially in the material products under acidic conditions.’ ‘The absorption spectrum of our materials correspond to 6-hydroxy-3-fluorone.citation’

I recommend the acceptance conditioning that authors modify their description of catalysts’ structure, following above-described arguments.

The above sentences were added to the manuscript (page 6) referring with new refs 37 and 38 to avoid misunderstanding of the readers. The word “oxonol structure” was inserted in pages 3 and 6 for clarity.